# Effects of Different Probiotics on the Volatile Components of Fermented Coffee Were Analyzed Based on Headspace-Gas Chromatography-Ion Mobility Spectrometry

**DOI:** 10.3390/foods12102015

**Published:** 2023-05-16

**Authors:** Linfen Zhao, Yanhua Wang, Dongyu Wang, Zejuan He, Jiashun Gong, Chao Tan

**Affiliations:** 1College of Food Science and Technology, Yunnan Agricultural University, Kunming 650201, China; zlfqcsj@163.com (L.Z.); 18487719653@163.com (Y.W.); 18849688676@163.com (D.W.); 18213305931@163.com (Z.H.); 2Agro-Products Processing Research Institute, Yunnan Academy of Agricultural Sciences, Kunming 650201, China; gong199@163.com

**Keywords:** coffee, probiotics, fermentation, aroma, headspace-gas chromatography-ion mobility spectrometry

## Abstract

Headspace-gas chromatography-ion mobility spectrometry (HS-GC-IMS) was used to study the effects of four kinds of probiotics on the volatile components of fermented coffee. The fingerprints showed that 51 compounds were confirmed and quantified, including 13 esters, 11 aldehydes, 9 alcohols, 6 ketones, 3 furans, 5 terpenes (hydrocarbons), 2 organic acids, 1 pyrazine, and 1 sulfur-containing compound. After fermenting, the aroma of the green beans increases while that of the roasted beans decreases. After roasting, the total amount of aroma components in coffee beans increased by 4.48–5.49 times. The aroma differences between fermented and untreated roasted beans were more significant than those between fermented and untreated green beans. HS-GC-IMS can distinguish the difference in coffee aroma, and each probiotic has a unique influence on the coffee aroma. Using probiotics to ferment coffee can significantly improve the aroma of coffee and provide certain application prospects for improving the quality of commercial coffee beans.

## 1. Introduction

Volatile flavor components in coffee are one of the key factors affecting beverage quality and a decisive consumer parameter. The quality of coffee can be affected by many factors, including growth conditions [1] and post-harvest treatment [2]. After harvesting, during this on-farm processing, the coffee cherries are subjected to a natural fermentation in which the coffee pulp is hydrolyzed by microbial growth [3].

Fermentation triggers an array of chemical changes within the beans that are precursors to volatile compounds formed during roasting [4]. Microorganisms play a key role in the formation of coffee flavor. Spontaneous fermentations have numerous disadvantages compared to fermentations with starter cultures. The structural and sensory characteristics of the product are improved using starter cultures, and the growing risk of harmful organisms can be prevented [5]. *Lactobacillus rhamnosus* has the potential to alter various flavor-related constituents in green coffee beans, which may lead to the modification of the final coffee flavor after roasting [6]. The inoculation of the *Lactobacillus plantarum* LPBR01 strain also significantly increased the formation of volatile aroma compounds during the fermentation process [7]. In addition, it has been found to have positive impacts on the coffee aroma when it optimizes the fermentation conditions and parameters [8]. For example, inoculates pectinolytic yeasts *Pichia guilliermondi* and *Candida parapsilosis* onto coffee cherries to facilitate mucilage removal [9].

In the present study, although several novel methods of coffee aroma modulation involving microbial fermentation have been reported, previous studies mainly attempted to modulate the volatile and aroma profiles of preformed coffee aroma extracts via microbial fermentation [10]. However, there have been few reports related to the use of different probiotics to enhance the flavor of commercial green coffee beans for specialty coffee. Therefore, there is a need to increase the value and improve the quality of Arabica commercial beans, which reduces the sensory differences and makes the flavor and beverage quality of Arabica commercial beans more similar to Arabica specialty beans, increasing the choice for consumers at a lower final product cost.

At present, solid phase microextraction-gas chromatography-mass (SPME-GC-MS), electronic nose and combination technology, comprehensive two-dimensional gas chromatography time-of-flight mass spectrometry (GC × GC/TOFMS), and gas chromatography-olfactometry (GC-O) are mainly used in flavor and aroma detection [11,12,13,14]. These detection technologies have a number of problems, including the gas sensor array drift phenomenon, being easily affected by environmental factors, complex pretreatment, and difficulty dealing with complex sample analysis [15]. Gas chromatography-ion mobility spectrometry (GC-IMS) has been demonstrated to be an effective, sensitive, rapid, and accurate analytical technique used for the identification and quantification of volatile organic compounds (VOCs) in the gas phase [16,17,18,19]. The advantage of this approach is that no sample preprocessing is required. Through headspace injection, the information of the trace VOCs in the sample can be quickly analyzed with a resolution as low as mug/m^3^ or even ng/m^3^ to form an intuitive fingerprint. Furthermore, GC-IMS allows the 2D separation of complex samples [20]. Based on the GC retention index (RI) and the migration time (Dt) of two-dimensional cross-qualitative [21], as well as the NIST database, with the gas chromatographic retention index, drift time, and peak volume, VOCs can be analyzed qualitatively and detected quantitatively.

Similarly, HS-GC-IMS has been used to identify the authenticity, freshness, shelf life, and variety of food samples. In recent years, HS-GC-IMS analysis has been extensively applied to investigate volatile compounds such as honey [22], kumquats [23], wine [24], goat cheese [25], Chinese material medica [26], frozen pork [27], and eggs [28]. Therefore, HS-GC-IMS can be used to establish the characteristic volatile fingerprint of probiotic fermented coffee. In this study, four kinds of probiotics were inoculated into Arabica coffee beans for fermentation. After fermentation, the objective of this study was to analyze the differences in volatile compounds between green coffee beans and roasted coffee beans using the HS-GC-IMS technique. Visual fingerprinting and matching matrix analysis were used to investigate the effect of differentiated probiotics on aroma during coffee fermentation, which provides a theoretical foundation and data support for this study of flavor changes in coffee fermentation.

## 2. Materials and Methods

### 2.1. Materials and Reagents

Coffee beans were purchased from Pu’er City, Yunnan Province, China. Arabica seed Catimor coffee beans were grown during the 2018 harvest season. Fermentation strains include *Lactobacillus acidophilus* CICC20244, *Bifidobacterium longum* subsp. *Infantis* CICC6069, *Streptococcus thermophilus* CICC20367, and *Lactobacillus paracasei* CICC20241, strains were purchased from the China Center of Industrial Culture Collection (Beijing, China).

Ultrapure water (18.2 MΩ cm^−1^, Milli-Q Plus system, Millipore, Bedford, MA, USA) was used throughout the work. Nitrogen gas with a purity of 5.0 was supplied by Newradar (Wuhan, China).

### 2.2. Instrumentation

Measurements were made using an HS-GC-IMS instrument (FlavourSpec^®^, Qingdao, China) in the G.A.S. Department of Shandong HaiNeng Science Instrument Co., Ltd. (Qingdao, Shandong, China). The device was also equipped with an automatic sampler unit (CTC-PAL, CTC Analytics AG, Zwingen, Switzerland), which improves the reproducibility of measurements. The ground coffee (1 g) was transferred into a headspace injection bottle (20 mL) and kept for 20 min (40 °C) with two replicates for HS-GC-IMS analysis. The top gas (200 μL) was analyzed by GC. GC was performed with a 15 m gas chromatographic column (FS-SE-54-CB-1, ID: 0.53 mm) to separate volatile components and couple them to IMS. A nitrogen of 99.999% purity was used as the carrier gas at programmed flow rates as follows: 2 mL·min^−1^ for 2 min, 2–15 mL·min^−1^ for 8 min, 20–80 mL·min^−1^ for 10 min, and 100–130 mL·min^−1^ for 20 min. The analytes were eluted and separated at 45 °C and then ionized in the IMS ionization chamber containing a tritium ionization source, which uses hydrated protons generated by the tritium source as its calibrant/reactant ion. An IMS electric field strength of 500 V/cm and a 150 mL·min^−1^ drift gas flow were applied in a drift tube of 98 mm length and operated at 45 °C. IMS was performed at ambient pressure.

### 2.3. Coffee Sample Preparation

Five kilograms of coffee cherries were washed three times in sterile, ultra-pure water, then put into a high-temperature cooking bag with a vacuum seal and sterilized under the conditions of 121 °C, 0.12 MPa, and 15 min. After sterilization, the samples were divided into 5 parts (A–E) and put into a sterile operation table, including non-treated coffee beans (A), coffee beans inoculated with *Lactobacillus acidophilus* (B), coffee beans inoculated with *Bifidobacterium longum subsp. Infantis* (C), coffee beans inoculated with *Streptococcus thermophilus* (D), and coffee beans inoculated with *Lactobacillus paracasei* (E). Each kilogram sample was subjected to inoculation with 5 mL of probiotic solution (≥1 × 10^6^ CFU/mL). The inoculated coffee beans were placed in a sterile fermentation bag with a one-way ventilation valve, and anaerobic fermentation was conducted at 18 °C for 48 h. After 24 h of fermentation, coffee cherries were rubbed to remove the skin in a fermenting bag, and the gas was expelled every three hours. Following fermentation, the samples were treated at 100 °C for 10 min, and screened out, and oven-dried in a thermostatic air-drying box at 35 °C until the moisture content was 12%. They were stored at room temperature in a one-way breathable bag before use. Non-treated (A) and treated green beans (B–E) were roasted with a SANTOKER R500E coffee roaster (Beijing Sandouke Technology Co., LTD., Beijing, China) under a constant air temperature of 185 ± 5 °C for 18 min to achieve a dark roast level. The agtron value of coffee beans is set at 35–40. Roasted samples were ground with an electronic coffee grinder (EK-43, MAHLKÖNIG, Hamburg, Germany) and then collected in a gas blocking packaging and vacuum.

The first letter of the sample name indicates the fermentation method, and the second letter indicates whether it was roasted or not (G for green beans and R for roasted coffee beans).

### 2.4. Statistical Analysis

All data were acquired and processed using the Laboratory Analytical Viewer (LAV) software (version 2.0.0, G.A.S., Beijing, China). GC-IMS Library Search software supplied by G.A.S. (Gesellschaft für analytische Sensorsysteme mbH, Beijing, China) was employed to identify unknown compounds. The software used the NIST and IMS databases to qualitatively analyze the components. IMS was an analytical technique for characterizing ionic chemical substances based on the difference in migration velocities of different gaseous ions in an electric field. According to the retention index (RI) and migration time (Dt), two-dimensional qualitative analysis is an excellent proposal for compound identification. The substances in the library can be qualitatively matched easily, which is more accurate than the degree of matching obtained by GC-MS.

The matching matrix was processed using Origin software version 9.1 (OriginLab, Northampton, MA, USA), and the data were presented as mean values ± standard deviation. Statistical analysis was conducted by SPSS software version 24.0 (SPSS Inc., Chicago, IL, USA). All samples were measured in duplicate.

## 3. Results and Discussion

### 3.1. Coffee Flavor Composition Analysis Based on HS-GC-IMS

The fingerprint shows that a total of 100 signal peaks were detected, of which 39 signal peaks were unknown compounds. Sixty-one signal peaks (51 compounds) were identified and quantified, including 13 esters, 11 aldehydes, 9 alcohols, 6 ketones, 3 furans, 5 terpenes (hydrocarbons), 2 organic acids, 1 pyrazine, and 1 sulfur-containing compound.

As shown in Table 1, the contents of the top five compounds in green coffee beans that have been treated differently are different.

The aroma components ranking in the top five in AG samples were acetone, dimethyl disulfide, hexanal, ethanol, and 5-methylfurfural. In BG samples, there were dimethyl disulfide, acetone, hexanal ethanol, and 3-methylbutanal. In CG samples, there were acetone, dimethyl disulfide, hexanal, 2-methylbutanal, and 3-methylbutanal. In DG samples, there were acetone, dimethyl disulfide, 2-methylbutanal, hexanal, and 3-methylbutanal. In EG samples, there were acetone, dimethyl disulfide, hexanal, ethanol, and 2-methylbutyric acid. The types and content of aromatic compounds in roasted coffee beans treated in different ways are essentially the same.

The aroma components ranked in the top five in the AR sample were furfural, 5-methylfurfural, acetone, furfuryl alcohol, and 2-butanone. In the BR sample were 5-methylfurfural, furfural, acetone, furfuryl alcohol, and 2-butanone. In the CR sample were 5-methylfurfural, furfural, acetone, 2-butanone, and furfuryl alcohol. In the DR sample were 5-methylfurfural, furfural, acetone, furfuryl alcohol, and 2-butanone. In the ER sample were 5-methylfurfural, furfural, acetone, furfuryl alcohol, and 2-butanone. In conclusion, from the total aroma components, the aroma content in green beans treated with probiotics (BG, CG, DG, EG) increased compared with the untreated one (AG), while the aroma content in roasted beans (BR, CR, DR, ER) decreased compared with the untreated one (AR); roasted coffee (R) has 4.48 to 5.49 times more aroma than green coffee (G).

### 3.2. Effects of Fermented Coffee with Single Probiotics on the Characteristic Components of Coffee Flavor

As shown in Figure 1, combined with Table 2, we observed the effects of the characteristic components of coffee flavor that were fermented with single probiotics. Firstly, comparing sample AG and sample BG, it was found that green beans fermented with *Lactobacillus acidophilus*, whose 15 signal peaks increased and 12 additional signal peaks decreased, which also significantly increased the content of ethyl acrylate (*p* < 0.05) while the content of 3-pentanone decreased, but the other strains did not have this ability. Secondly, focusing on samples AG and CG, we can draw the conclusion that green beans fermented with *Bifidobacterium longum* subsp. *Infantis*, whose 19 signal peaks increased significantly and 9 additional signal peaks reduced, also significantly increased the content of benzaldehyde, cyclohexanone, and propyl acetate-D, while the content of ethyl acrylate (as opposed to *Lactobacillus acidophilus*) and (E)-3-hexen-1-ol reduced. In addition, comparing samples AG and DG, it could be seen that green beans fermented with *Streptococcus thermophilus,* whose 18 signal peaks increased and 12 signal peaks decreased, which also significantly decreased the content of 2-methylpyrazine. Finally, comparing sample AG and sample EG, it was found that green beans fermented with *Lactobacillus paracasei,* whose 15 signal peaks increased and an additional 11 signal peaks decreased, which also significantly increased 1-heptanal, 4-methyl-2-pentanone-D, and 3-methyl-3-buten-1-ol while the content of cyclohexanone, hexanal-D, pentanal-M, and pentanal-D reduced.

Focusing on samples AR and BR, it was found that coffee beans fermented with *Lactobacillus acidophilus* after roasting had 8 signal peaks that increased and an additional 15 signal peaks that decreased, which also significantly increased the content of 2-methylbutan-1-ol content (*p* < 0.05). Through comparing samples AR and CR, it was found that coffee beans fermented with *Bifidobacterium longum* subsp. *Infantis* after roasting had 13 signal peaks that significantly increased and an additional 20 signal peaks that decreased, which also significantly increased the content of benzaldehyde and reduced the content of 5-methylfurfural-M, furfuryl alcohol, methyl acetate, 2-ethylfuran-M, and other compounds. Comparing sample AR and DR, it could be seen that coffee beans fermented with *Streptococcus thermophilus* after roasting, whose 8 signal peaks increased and an additional 20 signal peaks decreased, which also significantly increased 3-methylbutanol and ethyl 3-methylbutanoate-M. Through comparing samples AR and ER, it was found that coffee beans fermented with *Lactobacillus paracasei* after roasting had 12 signal peaks that increased and 15 signal peaks that reduced, with no significant difference between compound increase and decrease.

### 3.3. Common Effects of Fermented Coffee with Probiotics on Characteristic Components of Coffee Flavor

According to the fingerprints of all samples generated by the gallery plot in Figure 1, there were significant differences (*p* < 0.05) between untreated green beans (AG) and an additional four fermented samples (BG-EG). Furthermore, the signal peak intensity was enhanced by some probiotics while the other probiotics decreased it (39 signal peaks marked with “*”). No matter what kind of probiotics were used to ferment coffee, there were 12 signal peaks that changed, among which 6 signal peaks increased significantly, including dimethyl disulfide, 2-methylbutanal, 2-methyl butyric acid, ethyl 2-methylpropanoate-M, 4-methyl-2-pentanone-M, and propyl acetate-M; moreover, 6 signal peaks decreased (*p* < 0.05), including 2-ethylfuran-D, acetic acid, methyl acetate, ethyl 3-methylbutanoate-M, 2-ethylfuran-M, 2-propanol, and other compounds. Among the increasing compounds, dimethyl disulfide features a strong onion odor [29]; 2-methylbutanal features a powerful and choking odor [30], with peculiar cocoa and coffee flavors after diluting, and it was reported that the compound imparts chocolate, sweet, and fruity odors to coffee. The aroma was defined as intense but not typically cheese-like [31]; 2-methylbutyric acid features spicy qualities similar to roquefort cheese, which has a pleasant fruity aroma at low concentration; the aroma profile of ethyl 2-methylpropanoate-M features fruit and cream; 4-methyl-2-pentanone has green, herbal, and fruity characteristics [32]; and the aroma profile of propyl acetate features fruit such as pear and raspberry with some pleasant and bittersweet qualities. In addition, as to the decreasing compounds, 2-ethylfuran-M has a rubber-like and smoky burnt odor [33], with a warm and sweet flavor similar to coffee after diluting; acetic acid is an important odor-active compound having strong and pungent vinegar characteristics [34]; methyl acetate has a pleasant and slightly bitter flavor with an apple aroma [35]; and ethyl 3-methylbutanoate-M has strong vinous and liquor characteristics. In short, probiotics play an important part in the formation of flavor in green beans. Meanwhile, the changes in different signal intensity result in the production of a variety of coffee flavors.

Similarly, there were significant differences (*p* < 0.05) between untreated roasted beans (AR) and the other four fermented samples after roasting (BR-ER). Some probiotics increase the signal intensity of some compounds while making the others decrease (45 signal peaks marked with “*”). No matter what kind of probiotics were used to ferment coffee, there were 17 signal peaks that changed, among which 4 signal peaks increased significantly, including 2-pentanone, 2-butanone, heptanal, and 1-pentanol, and 13 signal peaks decreased (*p* < 0.05), including gamma-butyrolactone-D, 2-acetylfuran-M, furfural, 2-methylpyrazine, ethyl acrylate, 3-methylbutanal, ethyl acetate, acetone, ethyl propanoate, ethyl butanoate, 4-methyl-2-pentanone-M, 4-methyl-2-pentanone-D, and propyl acetate-M. Among the increasing compounds, 2-pentanone has a slight fruit odor; the aroma profile of 2-butanone features fragrant milk [36]; heptanal is related to the pungent, fishy, and unpleasant nutty odor; and 1-pentanol has a sweet and pleasant wax odor whose aroma profile features burning. Additionally, for those decreasing compounds, the aroma profile of gamma-butyrolactone-D featured faint, sweet, and aromatic buttery and milky [37]; 2-acetylfuran-M has the odor of sweet balsam, almond, cocoa, and caramel coffee [38]; The aroma profile of furfural characterizes almond, bitter, and toasted notes [39]; 2-methylpyrazine has nutty, cocoa, roasted, chocolate, popcorn, and coffee-like aroma [40]; ethyl acrylate emits rum smell; 3-methylbutanal features a powerful, acrid, pungent and choking odor, which was reported that the compound imparts fruity, fatty, animal and malty odors to coffee; ethyl acetate has strawberry and pineapple odor with pleasant ethereal fruity and brandy; acetone has a pleasant odor; ethyl propanoate has an odor reminiscent of rum and pineapple; ethyl butanoate has a fruity odor with pineapple and sweet taste. In a word, after fermenting with probiotics and roasting, the changes in decreasing signal peak intensities were more than those in increasing signal peak intensities. The differences in aroma between probiotic-fermented roasted beans and untreated roasted beans were more significant than those between probiotic-fermented green coffee beans and untreated green coffee beans.

### 3.4. Aroma Matching Matrix for All Samples

The greater the data match, the more coffee aroma will be similar to each other in the matching matrix. When the matching degree of two samples ranged between 90 and 100, the aromas in the two samples were extremely similar to each other and difficult to distinguish. As is shown in Figure 2, the aroma components of green beans fermented with *Bifidobacterium longum* subsp. *Infantis* (CG) were extremely similar to those of green beans fermented with *Streptococcus thermophilus* (DG), whose matching degree ranged between 85 and 90. Moreover, making a comparison between these 5 green beans, it was found that the aromas of samples BG, CG, and DG were similar to each other, whose matching degrees ranged between 90 and 95. There was a unique aroma found in sample EG whose matching degree ranged between 80 and 85 compared with other samples. In addition, there was a difference between green beans (AG) and additional fermented beans (CG-EG), and the matching degree ranged between 80 and 85.

Among the roasted samples, making a comparison between these 5 roasted beans, the matching degree of sample AR and other fermented beans (BR-ER) was less than 80, indicating that the aroma components of roasted beans changed significantly compared with untreated ones after fermentation. The aroma components of roasted beans treated with different probiotics (BR-ER) were similar to each other but slightly different, with matching degrees ranging between 80 and 90.

Moreover, comparing green and roasted beans, it could be seen that the matching degree of all samples was less than 40%, indicating that the aroma of coffee changed greatly after roasting.

## 4. Conclusions

The present study concluded that the effects of four different probiotics on fermented arabica coffee’s volatile components. The fingerprints by HS-GC-IMS showed that 61 signal peaks for 51 compounds were confirmed and quantified in all types of coffee, including 13 esters, 11 aldehydes, 9 alcohols, 6 ketones, 3 furans, 5 terpenes (hydrocarbons), 2 organic acids, 1 pyrazine, and 1 sulfur-containing compound. From the total aroma components, the aroma content in green beans treated with probiotics increased compared with untreated ones (AG), while the aroma content in roasted beans decreased compared with untreated ones (AR), which resulted in a 4.48–5.49-fold increase in aroma component contents after roasting. The differences in aroma between probiotic-fermented roasted beans and untreated roasted beans were more significant than those between probiotic-fermented green coffee beans and untreated green coffee beans. In conclusion, HS-GC-IMS can distinguish the difference in coffee aroma, and different probiotics have a unique influence on coffee aroma. Fermenting coffee with probiotics can significantly improve coffee aroma, which has excellent application prospects for improving the quality of commercial coffee beans.

## Figures and Tables

**Figure 1 foods-12-02015-f001:**
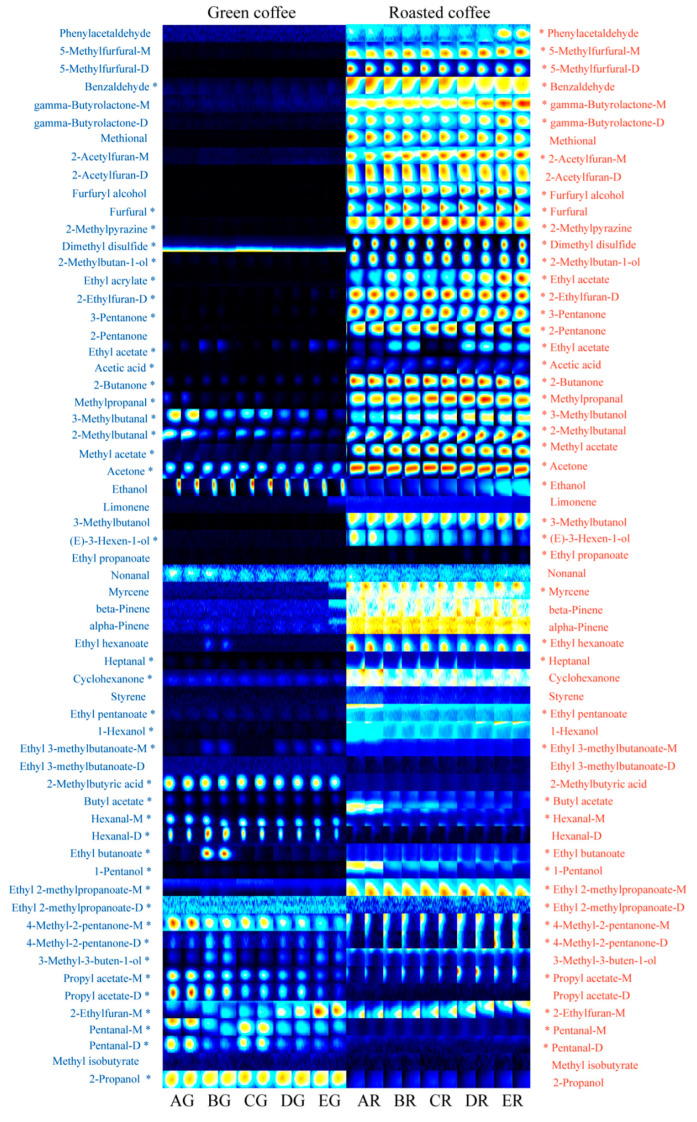
Gallery plot of coffee samples. “*”, represents that there were significant differences in signal peaks between untreated green and roasted beans (A) and another fermented sample (B or C or D or E) (*p* < 0.05). The box indicates there were significant differences in signal peaks between untreated green and roasted beans (A) and others fermented with different probiotics (B–E) (*p* < 0.05).

**Figure 2 foods-12-02015-f002:**
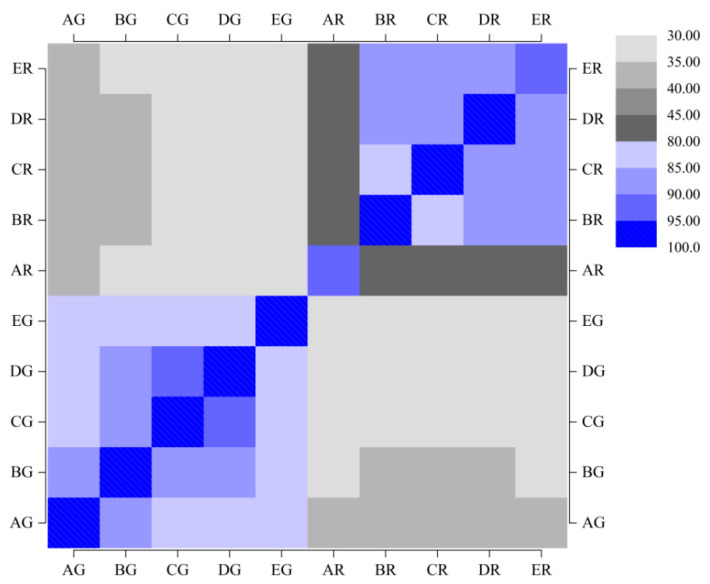
Aroma matching matrix for all samples.

**Table 1 foods-12-02015-t001:** Information and odour description of aroma compounds detected by HS-GC-IMS.

No.	Aroma Compound	Functional Group	CAS	RI	Rt [sec]	Dt [RIP rel]	^a^ Odour Description
1	Phenylacetaldehyde	aldehyde	122-78-1	1037.9	828.360	1.259	hyacinth, unpleasant, pungent, bitter flavor, sweet, fruit-like.
2	5-Methylfurfural-M	aldehyde	620-02-0	974.5	655.980	1.127	sweet, spicy, warm, caramel.
3	5-Methylfurfural-D	aldehyde	620-02-0	975.2	657.735	1.473	sweet, spicy, warm, caramel.
4	Benzaldehyde	aldehyde	100-52-7	961.4	623.805	1.145	sweet, oily, almond, cherry, nutty, woody.
5	gamma-Butyrolactone-M	ester	96-48-0	925.4	541.710	1.082	faint, sweet, aromatic, buttery, milky, creamy, peach.
6	gamma-Butyrolactone-D	ester	96-48-0	927.2	545.610	1.301	faint, sweet, aromatic, buttery, milky, creamy, peach.
7	Methional	aldehyde	3268-49-3	930.5	552.630	1.391	potato, musty, tomato, cheeses, onion, beefy brothy, egg, seafood.
8	2-Acetylfuran-M	furans	1192-62-7	916.0	522.210	1.120	sweet, balsam, almond, cocoa, caramel, coffee.
9	2-Acetylfuran-D	furans	1192-62-7	914.3	518.700	1.440	sweet, balsam, almond, cocoa, caramel, coffee.
10	Furfuryl alcohol	alcohol	98-00-0	865.3	430.170	1.307	burnt, sweet, caramellic and brown.
11	Furfural	furans	98-01-1	837.4	388.050	1.345	almond.
12	2-Methylpyrazine	pyrazine	109-08-0	839.8	391.560	1.395	nutty, cocoa, green, roasted, chocolate, meaty.
13	Dimethyl disulfide	sulfur-containing	624-92-0	734.0	262.470	0.986	diffuse, intense onion odor.
14	2-Methylbutan-1-ol	alcohol	137-32-6	778.3	311.415	1.473	cooked, roasted aroma with fruity or alcoholic undernotes.
15	Ethyl acrylate	ester	140-88-5	725.5	254.085	1.410	pungent odor.
16	2-Ethylfuran-D	furans	3208-16-0	714.2	243.750	1.330	smoky burnt, warm, sweet, coffee-like.
17	3-Pentanone	ketone	96-22-0	697.1	229.515	1.356	acetone-like odor.
18	2-Pentanone	ketone	107-87-9	684.6	220.350	1.373	sweet, fruity, banana.
19	2-Methylbutanal	aldehyde	96-17-3	659.7	204.555	1.401	fruity, musty with a berry nuance, musty, nutty, cereal, caramel.
20	3-Methylbutanal	aldehyde	590-86-3	646.0	197.145	1.409	acrid, pungent, apple.
21	Acetic acid	organic acid	64-19-7	626.6	187.590	1.324	sour pungent, cider vinegar, malty.
22	Ethyl acetate	ester	141-78-6	609.0	179.595	1.339	fruity, brandy, pineapple.
23	2-Butanone	ketone	78-93-3	588.8	170.820	1.248	sweet apricot.
24	Methylpropanal	aldehyde	78-84-2	560.0	158.340	1.283	a characteristic sharp, pungent odor.
25	Methyl acetate	ester	79-20-9	536.0	148.005	1.196	fruity, fresh, rum and whiskey
26	Acetone	ketone	67-64-1	510.8	137.085	1.119	pleasant odor.
27	Ethanol	alcohol	64-17-5	467.9	118.560	1.046	fruity.
28	Limonene	terpene	138-86-3	1017.6	770.835	1.216	sweet, orange, citrus, terpy.
29	3-Methylbutanol	alcohol	30899-19-5	742.6	271.245	1.338	fusel, fermented, fruity, banana, ethereal, cognac.
30	(E)-3-Hexen-1-ol	alcohol	928-97-2	859.1	420.420	1.529	grassy green aroma.
31	Ethyl propanoate	ester	105-37-3	709.7	239.850	1.452	rum, pineapple.
32	Nonanal	aldehyde	124-19-6	1105.2	1028.040	1.477	sweet waxy, orange.
33	Myrcene	terpene	123-35-3	992.6	702.780	1.219	terpy, herbaceous, woody, rosy, celery, carrot.
34	beta-Pinene	terpene	127-91-3	969.8	644.280	1.216	woody, piney, turpentine, minty, eucalyptus, camphoraceous.
35	alpha-Pinene	terpene	80-56-8	928.1	547.560	1.216	citrus, spicy, pine, turpentine.
36	Ethyl hexanoate	ester	123-66-0	1006.5	740.220	1.339	fruity, pineapple, banana.
37	Heptanal	aldehyde	111-71-7	901.2	492.960	1.333	pungent odor.
38	Cyclohexanone	ketone	108-94-1	891.2	474.240	1.150	peppermint, acetone.
39	Styrene	terpene	100-42-5	883.5	460.590	1.503	aromatic odor.
40	Ethyl pentanoate	ester	539-82-2	885.9	464.880	1.258	fruity.
41	1-Hexanol	alcohol	111-27-3	876.4	448.500	1.327	herbaceous, fragrant, mild, sweet, green fruity odor and aromatic.
42	Ethyl 3-methylbutanoate-M	ester	108-64-5	850.2	406.770	1.259	strong, fruity, vinous, apple.
43	Ethyl 3-methylbutanoate-D	ester	108-64-5	851.7	409.110	1.651	strong, fruity, vinous, apple.
44	2-Methylbutyric acid	organic acid	116-53-0	831.9	380.250	1.205	acidic sour, pungent, ripe fruit leather, lingonberry, dirty cheesy, fermented pineapple fruity.
45	Butyl acetate	ester	123-86-4	810.4	351.390	1.237	fruity, sweet, pineapple.
46	Hexanal-M	aldehyde	66-25-1	791.6	327.600	1.259	green, fatty, leafy, vegetative, fruity, woody.
47	Hexanal-D	aldehyde	66-25-1	791.6	327.600	1.559	green, fatty, leafy, vegetative, fruity, woody.
48	Ethyl butanoate	ester	105-54-4	795.3	332.085	1.206	fruity.
49	1-Pentanol	alcohol	71-41-0	766.0	296.985	1.251	intense fusel, fermented, bready.
50	Ethyl 2-methylpropanoate-M	ester	97-62-1	754.2	283.725	1.193	fruity.
51	Ethyl 2-methylpropanoate-D	ester	97-62-1	754.2	283.725	1.562	fruity.
52	4-Methyl-2-pentanone-M	ketone	108-10-1	733.8	262.275	1.176	sweet, ethereal, banana, fruity.
53	4-Methyl-2-pentanone-D	ketone	108-10-1	733.2	261.690	1.472	sweet, ethereal, banana, fruity.
54	3-Methyl-3-buten-1-ol	alcohol	763-32-6	733.6	262.080	1.247	NA
55	Propyl acetate-M	ester	109-60-4	711.1	241.020	1.163	fruity, pear, raspberry.
56	Propyl acetate-D	ester	109-60-4	709.9	240.045	1.476	fruity, pear, raspberry.
57	2-Ethylfuran-M	furans	3208-16-0	713.5	243.165	1.065	smoky burnt odor.
58	Pentanal-M	aldehyde	110-62-3	691.9	225.615	1.195	winey, fermented, bready, cocoa chocolate.
59	Pentanal-D	aldehyde	110-62-3	692.2	225.810	1.420	winey, fermented, bready, cocoa chocolate.
60	Methyl isobutyrate	ester	547-63-7	699.3	231.270	1.443	apple, pineapple, pricot.
61	2-Propanol	alcohol	67-63-0	515.3	139.035	1.179	unpleasant odor and a burning taste.

^a^ Odour description is taken from ChemicalBook database. M = monomer; D = dimer; RI = retention index; Rt = retention time; Dt = drift time; NA = no aroma description.

**Table 2 foods-12-02015-t002:** Effects of different probiotics on aroma composition of coffee.

Group Comparison	A vs. B		A vs. C		A vs. D		A vs. E	
Green coffee (G)	↑15	↓12	↑19	↓9	↑18	↓12	↑15	↓11
	2-Methylpyrazine	Furfural ^#^	Benzaldehyde *^#^	Ethyl acrylate *^#^	Dimethyl disulfide	Furfural ^#^	Dimethyl disulfide	2-Ethylfuran-D
	Dimethyl disulfide ^#^	2-Ethylfuran-D	2-Methylpyrazine	2-Ethylfuran-D	2-Methylbutanal	2-Methylpyrazine *^#^	2-Methylbutanal	Acetic acid
	Ethyl acrylate *	3-Pentanone *	Dimethyl disulfide ^#^	Acetic acid ^#^	3-Methylbutanal	2-Ethylfuran-D	Acetone	Ethyl acetate ^#^
	2-Methylbutanal	Acetic acid ^#^	2-Methylbutanal	Methyl acetate ^#^	Methyl propanal	Acetic acid ^#^	Heptanal ^#^	Methyl acetate
	3-Methylbutanal	Ethyl acetate ^#^	3-Methylbutanal	(E)-3-Hexen-1-ol *	Acetone	Ethyl Acetate ^#^	Ethyl pentanoate ^#^	Cyclohexanone *
	Methyl propanal	2-Butanone	Methyl propanal	Ethyl 3-methylbutanoate-M	Heptanal^#^	2-Butanone	1-Hexanol *	Ethyl 3-methylbutanoate-M
	Ethyl pentanoate	Methyl acetate	Acetone	3-Methyl-3-buten-1-ol	Ethyl pentanoate ^#^	Methyl acetate	2-Methylbutyric acid	Hexanal-D *
	2-Methylbutyric acid	Ethyl 3-methylbutanoate-M	Heptanal ^#^	2-Ethylfuran-M ^#^	2-Methylbutyric acid	Ethyl 3-methylbutanoate-M ^#^	Butyl acetate ^#^	2-Ethylfuran-M
	Hexanal-M	Ethyl 2-methylpropanoate-D	Cyclohexanone *	2-Propanol	Butyl acetate ^#^	ethyl 2-methylpropanoate-D	Ethyl butanoate	Pentanal-M *
	Hexanal-D	3-Methyl-3-buten-1-ol	2-Methylbutyric acid	-	Hexanal-M ^#^	3-Methyl-3-buten-1-ol	1-Pentano l^#^	Pentanal-D *
	Ethyl 2-methylpropanoate-M	2-Ethylfuran-M	Butyl acetate ^#^	-	Hexanal-D	2-Ethylfuran-M	Ethyl 2-methylpropanoate-M	2-Propanol
	4-Methyl-2-pentanone-M	2-Propanol	Hexanal-M ^#^	-	Ethyl butanoate	2-Propanol	4-Methyl-2-pentanone-M	-
	Propyl acetate-M	-	Hexanal-D	-	1-Pentanol ^#^	-	4-Methyl-2-pentanone-D *	-
	Pentanal-M	-	Ethyl 2-methyl-propanoate-M	-	Ethyl 2-methylpropanoate-M	-	3-Methyl-3-buten-1-ol *	-
	Pentanal-D	-	4-Methyl-2-pentanone-M	-	4-Methyl-2-pentanone-M	-	Propyl acetate-M	-
	-	-	Propyl acetate-M	-	Propyl acetate-M	-	-	-
	-	-	Propyl acetate-D^*^	-	Pentanal-M	-	-	-
	-	-	Pentanal-M	-	Pentanal-D	-	-	-
	-	-	Pentanal-D	-	-	-	-	-
Roasted coffee (R)	↑8	↓15	↑13	↓20	↑8	↓20	↑12	↓15
	Dimethyl disulfide	gamma-Butyrolactone-M	Phenylacetaldehyde	5-Methylfurfural-M *	2-Pentanone	gamma-Butyrolactone-D	Phenylacetaldehyde	gamma-Butyrolactone-M
	2-Methylbutan-1-ol *	gamma-Butyrolactone-D	Benzaldehyde *	gamma-Butyrolactone-D	2-Butanone	2-Acetylfuran-M	2-Pentanone	gamma-Btyrolactone-D
	2-Pentanone	2-Acetylfuran-M	Dimethyl disulfide	2-Acetylfuran-M	Methyl acetate	Furfural	2-Butanone	2-Acetylfuran-M
	2-Butanone	Furfural	2-Pentanone	Furfuryl alcohol *	Heptanal	2-Methylpyrazine	Methyl acetate	Furfural
	Methyl acetate	2-Methylpyrazine	2-Butanone	Furfural	Ethyl pentanoate	Ethyl acrylate	(E)-3-Hexen-1-ol	2-Methylpyrazine
	Ethyl hexanoate	Ethyl acrylate	(E)-3-Hexen-1-ol	2-Methylpyrazine	Butyl acetate	2-Methylbutanal	Ethyl hexanoate	ethyl acrylate
	Heptanal	3-Methylbutanal	Ethyl hexanoate	Ethyl acrylate	Hexanal-M	3-Methylbutanal	Heptanal	3-Methylbutanal
	1-Pentanol	Acetic acid	Heptanal	2-Methylbutanal	1-Pentanol	Acetic acid	Ethyl pentanoate	Ethyl acetate
	-	Ethyl acetate	Ethyl pentanoate	3-Methylbutanal	-	Ethyl acetate	Ethyl 3-methylbutanoate-M	Acetone
	-	Acetone	Ethyl 3-methylbutanoate-M	Acetic acid	-	Methyl propanal	Butyl acetate	Ethyl propanoate
	-	Ethyl propanoate	Butyl acetate	Ethyl acetate	-	Acetone	Hexanal-M	Ethyl butanoate
	-	Ethyl butanoate	Hexanal-M	Methyl propanal	-	3-Methylbutanol *	1-Pentanol	Ethyl 2-methylpropanoate-M
	-	4-Methyl-2-pentanone-M	1-Pentanol	Methyl acetate *	-	Ethyl propanoate	-	4-Methyl-2-pentanone-M
	-	4-Methyl-2-pentanone-D	-	Acetone	-	Ethyl 3-methylbutanoate-M *	-	4-Methyl-2-pentanone-D
	-	Propyl acetate-M	-	Ethyl propanoate	-	Ethyl butanoate	-	Propyl acetate-M
	-	-	-	Ethyl butanoate	-	Ethyl 2-methylpropanoate-M	-	-
	-	-	-	4-Methyl-2-pentanone-M	-	4-Methyl-2-pentanone-M	-	-
	-	-	-	4-Methyl-2-pentanone-D	-	4-Methyl-2-pentanone-D	-	-
	-	-	-	Propyl acetate-M	-	Propyl acetate-M	-	-
	-	-	-	2-Ethylfuran-M *	-	-	-	-

“*” represents that the probiotics can single-mindedly and significantly (*p* < 0.05) increase or decrease the content of the compound when green or roasted beans fermented with different probiotics were compared between these samples; “^#^” represents that the probiotics can single-mindedly and significantly (*p* < 0.05) increase or decrease the content of the compound when green or roasted beans fermented with the same probiotics were compared between these samples; “-” nothing; “↑” indicates the number of compounds with increased content. “↓” indicates the number of compounds with reduced content.

## Data Availability

Data is contained within the article.

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
