# Peer review of "Effects of Different Probiotics on the Volatile Components of Fermented Coffee Were Analyzed Based on Headspace-Gas Chromatography-Ion Mobility Spectrometry"

_foods, 2023, doi:10.3390/foods12102015_

Round 1
Reviewer 1 Report
Dear Authors,
your manuscript is of some interest, however, before it is published, it requires some revisions.
First of all along all the text and in the tables, there are several typos for example an uneven use of lowercase and uppercase letters. There are also some instances of errors related to the spacing between words and punctuation marks.
page 3 lines 140-142 and the following pages, tables and figure: I do not understand why in the indication of samples for green samples the letter F is used and for those roasted the letter B is much more immediate and understandable for readers to use the letters G and R. I ask the application of this replacement
Table 1:
The table requires some formatting. The formula column should be wide enough to accommodate formulas on a single line. if space has to be recovered the MW column can be deleted. also the function group. The column with the functional group can be deleted, it can be easily deduced from the chemical name. I suggest that for unrecognized compounds the "unknown" term be used not a number.
Page 4: the numerical results given in percentages would be better understood if presented in table form or through graphs
Section 3.2, tables 2 and figure 1:
This section is difficult to read and does not facilitate the understanding of the presented results. The table in particular is not very useful because graphically difficult to read and too dense of data
I suggest deleting the table and to comment the results only using the figure that should be improved because it seems of low resolution and also difficult to understand given the small size of the elements that constitute it
figure 2: the idea of presenting the results using this kind of representation is good but you have to give the inexperienced reader some more information about its meaning, the way the matching factor is calculated, and how to read the figure itself
Author Response
Dear Professor:
Thank you very much for your suggestion, I have made the changes as you suggested, please see the attachment
Thanks again for your help.

Reviewer 2 Report
In general, this paper presents interesting information, but the manuscript itself is not ready. Some points should be clarified and a revision is needed. The article is written carelessly when it comes to the editorial side. Please improve it. Focus on the quality and quantity of volatile compounds obtained during the experiment, not on their numerical equivalents
Introduction
l.32 correct (Vinicius de Melo Pereira, Soccol, Brar, Neto, & Soccol, 2017) Vinicius de Melo Pereira et al., 2017
l. 48 correct (Lee, Cheong, Curran, Yu, & Liu, 2016) Lee et al., 2016
l. 57 and 58 correct the citation
l 61, 64, 65 correct the citation
l.71 correct the citation
Please correct and standardize citation throughout the article
l.89-92 I suggest skipping it, because the purpose of the study was outlined earlier, and this already hooks into methodology
Results
l. 220 Lactobacillus acidophilus corrrect latin
l. 224 Bifidobacterium infantis corrrect latin
l. 235 add a legend below table 2
l. 166 onwards Acetone(16.40 ± 166 0.06%), Dimethyl disulfide(14.55 ± 0.47%), hexanal(3.61 ± 0.08%), ethanol(3.37 ± 0.02%), 167 5_Methylfurfural (2.15 ± 0.03%) I suggest not submitting these numerical results and leave the names of the obtained compounds alone
Author Response
Dear Professor:
Thank you very much for your suggestion, I have made the changes as you suggested, please see the attachment again
Thanks for your help.

Round 2
Reviewer 1 Report
Although my suggestions were only partially accepted the manuscript in its current version appears improved as a whole therefore I consider it worthy of publication
Author Response
尊敬的教授,非常感谢您的建议,我根据您的建议进行了修改,
可以在稿件中查看。
再次感谢您的帮助。如果您有任何疑问,请随时与我们联系。